# Perspectives and Tools in Liver Graft Assessment: A Transformative Era in Liver Transplantation

**DOI:** 10.3390/biomedicines13020494

**Published:** 2025-02-17

**Authors:** Kawthar Safi, Angelika Joanna Pawlicka, Bhaskar Pradhan, Jan Sobieraj, Andriy Zhylko, Marta Struga, Michał Grąt, Alicja Chrzanowska

**Affiliations:** 1Department of Biochemistry, Medical University of Warsaw, 02-097 Warsaw, Poland; kawthar.safi@wum.edu.pl (K.S.);; 21st Chair and Department of Cardiology, Medical University of Warsaw, 02-097 Warsaw, Poland; 3Department of General, Transplant and Liver Surgery, Medical University of Warsaw, Banacha 1A, 02-097 Warsaw, Poland

**Keywords:** living donor liver transplantation (LDLT), deceased donor liver transplantation (DDLT), hepatic steatosis (HS), transient elastography (TE), liver stiffness (LS), controlled attenuation parameter (CAP), marginal donors, donors after circulatory death (DCD), donors after brain death (DBD)

## Abstract

Liver transplantation is a critical and evolving field in modern medicine, offering life-saving treatment for patients with end-stage liver disease and other hepatic conditions. Despite its transformative potential, transplantation faces persistent challenges, including a global organ shortage, increasing liver disease prevalence, and significant waitlist mortality rates. Current donor evaluation practices often discard potentially viable livers, underscoring the need for refined graft assessment tools. This review explores advancements in graft evaluation and utilization aimed at expanding the donor pool and optimizing outcomes. Emerging technologies, such as imaging techniques, dynamic functional tests, and biomarkers, are increasingly critical for donor assessment, especially for marginal grafts. Machine learning and artificial intelligence, exemplified by tools like LiverColor, promise to revolutionize donor-recipient matching and liver viability predictions, while bioengineered liver grafts offer a future solution to the organ shortage. Advances in perfusion techniques are improving graft preservation and function, particularly for donation after circulatory death (DCD) grafts. While challenges remain—such as graft rejection, ischemia-reperfusion injury, and recurrence of liver disease—technological and procedural advancements are driving significant improvements in graft allocation, preservation, and post-transplant outcomes. This review highlights the transformative potential of integrating modern technologies and multidisciplinary approaches to expand the donor pool and improve equity and survival rates in liver transplantation.

## 1. Introduction

Orthotopic liver transplantation remains an evolving field in modern medicine and surgery and continues to gain importance as a life-saving treatment modality in the 21st century. First performed in humans 60 years ago, it is now the standard of care yet remains a complex and resource-demanding therapy for patients, healthcare providers, and society at large [1]. With liver disease rates globally on the rise, claiming the lives of nearly two million individuals worldwide each year [2], ongoing research to optimize graft assessment and utilization is crucial, especially in the context of today’s organ shortage [3]. The last two decades have witnessed a significant increase in graft demand but were not met by a paralleled increase in organ availability, resulting in increased waitlist mortality. American reports alone reveal that a serious lack of donor organs contributes to the deaths of over 2500 End-Stage Liver Disease patients annually [4]. Furthermore, results from retrospective donor data revealed that 15% of livers initially deemed non-transplantable could have been successfully used, emphasizing the need for refined graft assessment criteria [5]. Given this, and the lack of clear guidelines from organizations like the National Institute for Health and Care Excellence (NICE) around when to use marginal donor grafts (grafts considered not suitable for transplant without normothermic perfusion, older donors, or donors with steatotic livers), many transplant centers around the world are increasingly adopting a more flexible approach when considering the utilization of discarded liver grafts [5,6]. The New England journal of Medicine identified two primary considerations in the graft evaluation: urgency (on the basis of prognosis without transplantation) and barriers hindering a successful long-term outcome post-operatively [1]. Approaches to prioritizing patients vary internationally. Thus, in the United States, patients awaiting grafts from deceased donors are prioritized in accordance with the federal final rule for transplantation, which dictates donor allocation on the basis of the highest urgency. The Model for End-Stage Liver Disease (MELD) score is the main tool employed to assess this. MELD score is a clinical scoring tool used to evaluate chronic liver disease severity and prioritize waiting list candidates. It is based on laboratory values of bilirubin, creatinine, and International Normalized Ratio (INR), with higher scores indicating higher priority. In contrast, the United Kingdom (UK) shifted to a system allocating deceased-donor livers in relation to the anticipated patient benefit in 2018 [1]. With such frameworks in place, the US system can better offer timely transplants in emergencies in part due to its larger networks, while the UK attaches greater importance to more equitable and accessible resource distribution. In terms of outcomes, both countries share comparable success rates [7]. Currently, both systems are evolving to improve survival rates and transplant equity in light of recent innovations, such as normothermic perfusion, which helps improve organ preservation and recovery.

Common indications for liver transplantation include end-stage liver disease, metabolic disorders, cirrhosis, chronic fibrotic liver disease, selected hepatic cancers, and liver dysfunction symptoms that are refractory to non-surgical treatments [8]. Acute liver failure constitutes less than 5% of cases necessitating liver transplantation [1]. Unresolving portal hypertension in the context of other liver pathologies may also constitute an indication for this treatment option. Liver transplants may be classified as orthotopic, involving the replacement of the entire liver, or living donor transplants, which utilize partial liver grafts. Contraindications to transplant include uncontrolled infections, active substance abuse, and certain malignancies, as they are perceived to increase the risks to the recipient. Graft rejection, infections, and biliary complications, which may require lifelong immunosuppression and monitoring, are typical post-operative complications [8]. New technologies such as hepatocyte transplantation and extracorporeal liver perfusion offer hope in bridging patients to transplant by temporarily supporting hepatic function, although further research is essential to ensure their safety and efficacy.

In light of these recent advances in graft assessment—including scoring systems, biomarkers, imaging, and artificial intelligence—this review aims to explore how graft evaluation may be harnessed to expand the donor pool and optimize clinical outcomes, ultimately leading to more economical healthcare resource distribution. This discussion is structured around the clinical journey of both the donor (living and deceased) and recipient, beginning with the pre-donation testing stage, followed by assessments during the organ procurement period, and concluding with post-operative evaluation whilst incorporating the use of artificial intelligence. Detailed discussions of the differences and similarities in graft assessment between living and deceased donors are beyond the scope of this work. Instead, the review takes a more holistic discussion, with equal emphasis on recipient factors and donor quality.

## 2. Pre-Donation Assessment Phase

### 2.1. Pre-Donation Testing: General Overview

Initial liver graft function evaluation is a comprehensive process with its early stages taking place when the recipient is in the intensive care unit (ICU) in the case of a deceased donor [9,10]. Clinician assessment relies on donor and recipient history and preoperative data, clinical parameters, laboratory values, radiological studies, and histopathological findings. This assessment includes basic hematological and biochemical tests, serology for infection screening (HBsAg, anti-HCV, HIV, VDRL, and IgG anti-HBc), and an abdominal ultrasound with additional imaging. Female donors of childbearing potential undergo a pregnancy test. Ultrasound helps exclude structural abnormalities, and triple-phase computerized tomography (CT) of the abdomen provides information about liver steatosis and vascular anatomy, as well as information about volumetric status. Additionally, donor age, blood group, and any known history of steatosis are noted. The same serology tests are carried out for the recipient prior to the transplant [9,10].

### 2.2. Serological and Biochemical Tests

Commonly performed routine blood tests are complete blood count (CBC), liver function test (LFT), prothrombin time (PT), activated partial thromboplastin time (aPTT), as well as hepatitis B surface antigen (HBsAg), anti-hepatitis B core antigen (HBc), anti-hepatitis C virus (HCV), anti-HIV, and anti-cytomegalovirus (CMV) antibodies, among others [10]. In cases involving a deceased donor who has been hospitalized for more than 72 h prior to death, blood and urine culture tests are performed to exclude transmittable infections. The deceased donor can then be prepared for organ retrieval. During this time, a visual inspection of the potential liver graft is performed by a team of surgeons, and if deemed necessary, a liver biopsy is obtained. Pre-transplant liver biopsies provide information on the degree of steatosis where surgeons are doubtful and help identify severe fibrosis, which may be a contraindication for transplant [11].

#### 2.2.1. Biochemical (Static) Tests (STF)

LFTs are standard for preoperative surgical planning and include aspartate transaminase (AST) and alanine transaminase (ALT), which measure the liver’s functional reserve [9]. However, these tests are limited in that they are non-specific to liver changes but may also be influenced by the deceased donor’s health state near the time of death [10]. For example, AST levels may be elevated due to acute injury of cardiac or skeletal muscles, reflecting physical health changes in the donor rather than in the liver graft itself. Additionally, as they are easily influenced by factors like bleeding, they are unable to provide reliable information about direct liver function [12]. Furthermore, derangement in LFTs is common in brain-dead donors due to steatosis, sepsis, and hemodynamic instability, creating uncertainty about future graft function [9]. A retrospective cohort study of the United Nations Organ Sharing (UNOS) database revealed that there was no associated risk with using grafts from donors with peak ALT levels. This may be a door of opportunity to help expand the donor pool [13]. It should be emphasized that bilirubin, liver enzymes, and lactate levels, in relation to sepsis, cardiopulmonary resuscitation, or surgery progression, are the key deciding factors in determining if the graft can be used rather than relying on any single parameter [11,14].

#### 2.2.2. Liver Function Tests (LFTs)

Complementary to the commonly used biochemical (static) tests, such as LFTs, are hepatic function (dynamic) tests and scoring systems that may be used at the patient’s bedside. These tests play a crucial role in cases where cirrhosis and fibrotic liver disease wish to be excluded. They evaluate various physiological roles of the liver, including secretory, excretory, and metabolic functions, as well as detoxification and synthetic and selected enzyme functions [15]. Examples relevant to donors and recipients include the “quantitative liver function tests” proposed by Jochum et al., which consist of three pharmacokinetic tests: (1) Galactose elimination capacity (GEC), which acts as a marker of cytosolic capacity, (2) indocyanine green half-life (ICG), which informs about hepatic blood flow and intrahepatic bile excretion, and (3) lidocaine half-life which provides information on the functioning of the cytochrome P450 system and the detoxification ability of hepatocytes [16]. However, with the development of scoring tools such as L-GrAFT and EASE score, discussed later, the majority of dynamic tests are quickly losing popularity and have become largely abandoned [17].

One dynamic test that remains relevant to modern studies is ICG clearance. The indocyanine green (ICG) clearance test is a quantitative assessment of liver function using the amphiphilic carbocyanine molecule, indocyanine green, which is taken up by hepatocytes and excreted unchanged through bile [17]. ICG clearance at physiological levels verifies sufficient hepato-splanchnic blood flow, the existence of undamaged functioning hepatocytes, and unobstructed bile secretion [17,18]. Indocyanine green is administered intravenously with a dosage of either 0.25 mg/kg or 0.5 mg/kg. The ICG clearance test can then predict liver function during the early postoperative period and reflect the state of the splanchnic circulation [19]. The ICG normal half-life is 3–5 min, and physiologic ICG clearance occurs at a rate of 6–12 mL/min/kg [20]. Pulse dye densitometry, along with blood concentration measurement and imaging, are taken for a holistic evaluation of results. Data may be collected anytime between 6 h to 7 days postoperatively. ICG clearance has been used for decades to assess liver function, but due to its inconvenience in the clinical setting, it is rarely used nowadays. Instead, a more clinically accessible and noninvasive ICG clearance determination technology known as a liver function monitoring system (LiMON) provides a physiologic monitor to evaluate liver function after transplantation and in the ICU [21]. This technique is described in greater detail in the ex vivo section.

### 2.3. Imaging Techniques

After ensuring the donor’s fitness to undergo liver transplant surgery, imaging is required for liver volumetry assessment as well as to study the vascular and biliary anatomy [11]. Imaging includes abdominal CT and MRI with contrast, along with Magnetic Resonance Cholangiopancreatography (MRCP). Homogenous enhancement of all liver segments on CT identifies functional hepatic tissue. The same effect is achieved by MRI with a hepatobiliary contrast agent.

Conventional bedside ultrasonography plays a pivotal role in determining liver steatosis, primarily through echogenicity assessment. It is most effective in identifying moderate to severe steatosis, where increased echogenicity serves as the hallmark of fatty liver disease [22]. However, at the mild steatosis level (<30%), the sensitivity and specificity decrease significantly, with approximately 60–65% sensitivity for steatosis greater than 5%. In such instances, confirmation through complementary methods such as CT, MRI, and biopsy becomes necessary [22]. Moderate and severe steatosis, on the other hand, are more readily detected, with sensitivities and specificities above 90% [23]. Comparing liver echogenicity to that of surrounding structures like the renal cortex and the spleen can further aid in identifying moderate to severe steatosis [22]. For example, the hepatorenal index (HRI) is a semi-quantitative measure of the ratio of brightness between the liver and the right renal cortex in grayscale ultrasound. HRI has demonstrated an effective detection rate for mild steatosis, with higher accuracy for moderate and severe cases [24]. Ultrasound also aids in vascular structure, biliary anatomy, and liver morphology assessment, helping identify potential contraindications for transplantation, such as the presence of a hepatic adenoma [25]. Although not without limitations, ultrasound remains a first-line imaging modality in the evaluation of donor livers. Integrating other imaging tools would serve to provide a more holistic and thorough assessment of a graft’s suitability for transplantation.

An unenhanced CT scan is useful for hepatic steatosis (HS) assessment through the calculation of the liver attenuation index (LAI). LAI is calculated by selecting a region of interest of at least 1 cm^2^ in multiple places on the liver and spleen and calculating the difference between the mean hepatic and splenic attenuation [26]. LAI values of −10 to 5 HU suggest mild to moderate steatosis (6–30%), while values below −10 HU (−11, −12, … and so on) suggest moderate to severe hepatic steatosis (>30% fat). Values between 5 and 15 are considered acceptable, whereas negative values numerically smaller than −10 (−11, −12, −13, etc.) require further assessment by liver biopsy [27]. Given the risks associated with liver biopsy (pain, bleeding) and inconsistent results due to the localized nature of steatosis, other non-invasive methods, such as transient elastography (TE) and other imaging modalities, are increasingly preferred [28]. The hepatic artery is then examined, and its anatomy may be classified in accordance with one of the 10 subtypes of the Michel classification [29]. Liver volumetry is calculated using a standard liver volume formula or through MRI-derived volume estimation using artificial intelligence, which is discussed later. Next, the hepatic veins are reviewed with an emphasis on the middle hepatic vein location. The venous drainage of segments 4a and 4b and any variations are noted and usually described in the radiological report. Portal vein branches are also visualized to rule out thrombosis or signs of portal hypertension. Finally, MRCP is performed, particularly examining whether the right anterior lobe and right posterior lobe biliary ducts directly drain into the left lobe, which would contraindicate left lobe donation in partial graft transplants. The same applies if the left lobe biliary ducts drain into the right anterior and posterior lobes due to the risk of biliary complications.

### 2.4. Role of Transient Elastography

Transient elastography (FibroScan, TE) is a convenient and reproducible method that objectively measures fibrosis and steatosis by assessing liver stiffness (LS) and the controlled attenuation parameter (CAP). It relies on the principles of shear-wave and echo-degeneration, respectively [30]. Numerous studies have demonstrated that FibroScan provides high accuracy in assessing liver fibrosis and steatosis in patients with chronic liver diseases of various causes. However, research on its use in liver grafts from deceased brain donors is limited [31]. Thorough evaluation of hepatic steatosis prior to surgery is crucial for enhancing the recipient’s prognosis, reducing the need for liver retransplantation by optimizing liver graft quality, and increasing the availability of liver grafts by preventing the unnecessary rejection of suitable candidates [32]. The LS measurement shows a strong positive correlation with the fibrosis stage of the liver graft (r = 0.73, *p* < 0.01), while CAP is linked to the steatosis stage (r = 0.64, *p* < 0.01). Liu et al. concluded that TE is expected to be a useful tool for quantitatively assessing the level of HS in grafts from brain-dead donors [32]. When used alongside laboratory tests and ultrasonography, CAP offers the transplantation team additional insights regarding hepatic steatosis before liver procurement [32]. Further studies are investigating its role in postoperative liver graft evaluation.

During post-transplant follow-up, identifying increased LS with TE can help predict fibrosis development. To analyze the mechanical properties of tissues, a stimulus (stress) is applied, and the resulting deformation (strain) in the tissue is measured [33]. In chronic liver disease, there is an abnormal rise in the fibrosis component (collagen) within the liver, allowing measurements of tissue stiffness to act as indicators of the extent of fibrosis. The staging of fibrosis is the primary indication for elastography in hepatic imaging [34,35]. However, it is important to note that relying exclusively on elastography can be deceptive. Graft-related complications such as ischemia from a hepatic artery or portal vein thrombosis, biliary obstruction, venous outflow issues, and fluid collections around the graft can affect LS measurements. These factors can contribute to increased stiffness in the graft’s parenchyma, so it is essential to consider the entire clinical context [36]. For this reason, combining TE with MRI or ultrasound with Doppler may help improve overall graft assessment [36].

TE coupled with MRI-based techniques is an emerging alternative for HS assessment. These techniques include magnetic resonance elastography (MRE) and magnetic resonance imaging-estimated proton density fat fraction (MRIPDFF), which are unique in that they correlate with liver biopsy results. 107–110 MRI PDFF is also more accurate than ultrasound-based CAP measurement (AUROC of 0.85 for CAP vs. 0.99 for MRIPDFF) [37]. MRI PDFF uses the phase and magnitude of the MRI signal to accurately deduce the fat fraction in tissues by correcting for factors influencing signal intensity [38,39]. As a non-invasive examination, MRI-PDFF is quickly becoming one of the preferred modes of steatosis evaluation in most transplant facilities. MRE, on the other hand, is particularly useful for fibrosis assessment in the donor liver [36].

### 2.5. Liver Biopsy

Liver biopsy is the standard invasive procedure for histological analysis to exclude graft fibrosis, steatosis, or other underlying pathology. The most common liver quality evaluation method worldwide is wedge biopsy, which is taken during procurement. A wedge biopsy of 1 cm in side length is recommended to be performed in at least two segments, as supported by Frankel et al. [40,41]. It is a valuable tool for identifying and classifying steatohepatitis and assessing its progression through histological analysis. Histological examination is the definitive method to distinguish steatosis from steatohepatitis. This is crucial because the former may not impact the decision to use the graft, whereas the latter contraindicates it [42]. HS is the primary factor leading to the rejection of grafts, and liver biopsy remains the gold standard for quantitative and histological assessment. Depending on the fat accumulation extent, HS is classified as mild (<30%), moderate (30–60%), or severe (>60%). A biopsy can also reveal preservation injury, recurrence of native hepatic disease, rejection, viral infection, and biliary obstruction. It is the most useful test for the differential diagnosis of graft dysfunction and is routinely performed in cases where this is suspected, for instance, in unexplained clinical findings or persistently abnormal LFTs.

Additionally, biopsy allows for the use of the Meta-analysis of Histological Data in Viral Hepatitis (METAVIR) scoring system, which assesses the level of inflammation and fibrosis in patients with hepatitis C. In liver transplantation, it is useful for the identification of significant fibrosis, starting from stage F2 and above [43]. Similarly to METAVIR, the Hepascore biochemical scoring system, based on LFTs, predicts the severity of liver fibrosis or cirrhosis in hepatitis C infection [5]. The Hepascore can be used alongside the METAVIR score to assess hepatic fibrosis more accurately in liver grafts. Conversely, liver biopsy is not without limitations. As mentioned earlier, issues with accuracy and patient-related complications exist in cases of live donors. It is also invasive, expensive, time-consuming, and not always practical during liver procurement. Significant disparity still exists between surgical visual assessment and histology results, with one study reporting approximately 30% false negative histology results [40]. HS remains a unique challenge in liver transplantation, with ongoing debate around the best method of its evaluation. Currently, surgeon expertise and visual inspection (macroscopic examination) are the main methods in determining whether a biopsy for HS assessment should be performed.

Visual inspection by surgeons has poor sensitivity and specificity, even in cases of severe fatty infiltration [6,44] and cannot be standardized nor critically assessed for its ability to improve graft evaluation, particularly in terms of survival and function [45]. A double-blind evaluation of 201 donor livers, using donor criteria such as gross liver inspection and routine LFTs (of hospitalized donors), concluded that these current approaches are frequently unreliable and irreproducible as well as costly and uneconomical [45]. Consequently, the search for real-time, direct liver tissue assessments to quantify graft function and the risk of graft failure continues. In conclusion, while biopsy remains the gold standard to diagnose HS, autoimmune hepatitis, and other liver disorders that must be excluded, alternative methods, such as the previously discussed TE, are being developed to predict disease recurrence and HS grade. These advancements are crucial for expanding the organ donor pool. Another issue in using steatotic grafts is the discussion about which HS grafts are acceptable and for whom. According to the European Association for the Study of the Liver (EASL) clinical practice guidelines, grafts with >30% macrosteatosis are acceptable only with risk adjustment, meaning with a maximum Balance of risk (BAR) score of 9 at most [46,47]. BAR score is a simple scoring system of six donor and recipient variables to predict post-transplant patient survival [48]. More information on the BAR score and its limitations are found in the Table in Section 3.6.

An additional yet intriguing way that biopsies may assist in graft eligibility assessment is through metabolome studies using liquid chromatography-mass spectrometry (LC-MS). One clinical trial examining biopsies from DCD and DBD livers found that specific metabolite contents could stratify six-year survival outcomes (*p* = 0.0073) [49]. The study concluded that LC-MS detection of purine metabolites, combined with ALT at the pre-transplantation stage, could improve the prediction of early graft function and long-term survival. The ratios of AMP/urate, adenine/urate, and hypoxanthine/urate with ALT demonstrated the highest diagnostic accuracy (84%) for predicting transplant outcomes, with a confidence interval of 71% to 96%. Purine pathway metabolites play multiple roles in the biochemical and tissue level changes anticipated during graft preparation and transplantation, including regulating inflammation, signaling oxidative injury, and serving as markers of cell death. During periods of cold and warm ischemia, for example, their dysregulation is associated with inflammation, energy imbalances, and ischemic tissue damage. These findings highlight the potential of purine ratio analysis for prognosis prediction when juxtaposed with clinical data and donor information. Notably, they also presented superior predictive ability compared to conventional liver function tests. Researchers suggest that a test for purine metabolite analysis may eventually be performed intraoperatively using rapid evaporative ionization mass spectrometry, with the power to expedite decision-making during the back-table procedure stage. With a short turnaround time, this test could offer new avenues for optimizing graft selection and transplant outcomes [49]. While further validation and data from multicenter studies are required, metabolome analysis continues to be an exciting and unpredictable arena within the liver transplantation world.

A succinct overview of the steps in liver transplant function assessments is depicted in Figure 1 below.

### 2.6. Additional Examinations

Apart from liver examinations, other investigations, such as chest radiographs, electrocardiograms, and echocardiography, are also performed in all patients. The International Liver Transplant Society (ILTS) recommends a cardiovascular examination (including routine echocardiography) and, if required, stress echocardiography and/or coronary angiography. Screening for asymptomatic inherited coagulation disorders is also advised [51]. A thorough psychiatric evaluation is necessary prior to organ donation. Informed consent must be obtained after explaining the surgical, medical, and psychological risks of a hepatectomy [11].

## 3. Donor Qualities and Donor-Recipient Matching

### 3.1. Donor Types and Graft-Related Considerations

Graft evaluation remains a critical step for optimal prognosis and graft function in both deceased donor liver transplantation (DDLT) and living donor liver transplantation (LDLT). Donors may be broadly classified into either a deceased donor or a living donor. A deceased donor implies that the graft was isolated after brain death and, as such, is known as a donor after brain death (DBD). The majority of cadaveric donors satisfy these criteria, which include irreversible loss of brainstem function and reflexes accompanied by a non-responsive coma and no spontaneous ventilation [52]. Due to the scarcity of such donors, the second category of deceased donors is those after circulatory death (DCD) has been created and is accepted by several countries. In this group, cardio-respiratory criteria constitute death, and donors may be sub-categorized into controlled or uncontrolled DCD. Controlled DCD includes cases where withdrawal of life support was planned, whereas uncontrolled DCD is cases of death by sudden cardiac arrest due to either failure of return of spontaneous circulation, a patient’s wish to not be resuscitated, or patients being too frail to survive resuscitation attempts [53]. Very few grafts are isolated from uncontrolled DCD. Non-heart-beating donors may also be classified in more detail under the Maastricht donor categories [54].

An essential point to consider in graft preparation here is that, in uncontrolled DCD, death is unexpected, and organ retrieval time is naturally longer. In controlled DCD, however, withdrawal of ventilation is elective in end-of-life situations, allowing for more time to prepare for graft retrieval, thus limiting ischemia time [11]. Because of this, the main challenge with DCD, as a whole, is limiting warm ischemia time (WIT). WIT is defined as the time from the cessation of the heartbeat to the initiation of the organ preservation procedure [55]. As a result, DCD grafts, particularly uncontrolled ones, are more prone to the risk of primary non-function, preservation injury, delayed graft function, and biliary stricture. Rapid femoral/iliac artery cannulation for extracorporeal perfusion (ECMO) has been proposed as a measure to limit ischemia time [56]. More novel techniques, like the use of ex vivo machine perfusion, discussed in the post-transplantation section, are promising strategies to mitigate ischemia time and avoid static cold storage, thereby optimizing the procurement procedure to improve transplant outcomes [11]. Enhanced donor-recipient matching and a greater understanding of hemodynamic variables during life support withdrawal have contributed significantly to improvements in the utilization of DCD livers [57].

### 3.2. Living Donor Liver Transplantation (LDLT) Versus Deceased Donor Liver Transplantation (DDLT)

LDLT offers numerous benefits over DDLT, including greater organ availability and the ability to plan transplants electively, which can be lifesaving in acute liver failure, where time is crucial [58]. In this way, LDLT decreases waiting list mortality. Grafts obtained through LDLT tend to be of better quality due to minimal to no preservation injury, which is a common concern in DDLT grafts [59]. LDLT, however, comes with significant challenges, including complex surgical requirements, ethical and legal considerations, and risks to the living donor. The mortality risk for the donor is approximately 0.3–1% [60]. On the other hand, DDLT is more common among Western countries, and LDLT remains the popular option in Asia. More than 90% of transplants in the latter are LDLT [11]. This stems from differences in culture, the nature of family relations, and religious beliefs. The factors in graft evaluation of both donor types differ. In living donors, factors such as age (18–60 years), comorbidities, donor obesity, and any medical condition posing perioperative risks all need to be considered [61]. This is not the case with deceased donors, where there is a greater degree of flexibility in graft utilization and allocation [62]. Despite these differences, graft survival rates and patient outcomes for both types of donation are relatively similar, though ischemic cholangiopathy (IC) is more common in the DCD group [63]. Table 1 summarizes factors influencing graft function based on the evidence collected within the last 5 years.

### 3.3. Donor-Recipient Matching and the Expansion of the Donor Pool

One cannot provide a complete discussion of liver transplantation without addressing donor-recipient matching. The success of liver transplantation (LT) is greatly influenced by effective donor-recipient (D-R) matching, a process that determines the suitability of grafts. Certain patterns of selecting particular recipients to match with marginal donors are beginning to evolve. Many transplant centers are now attaching more importance to the use of “marginal donors”, which have been traditionally underutilized [72]. This is a relatively new graft classification, denoting grafts meeting extended criteria. Marginal grafts include those from higher-risk donors, such as those who are obese, over the age of 50, have macrovesicular steatosis exceeding 50%, have a history of ICU admission, or have extended cold ischemia times (>14 h) [73]. Recent advancements in perfusion methods have now deemed these grafts increasingly viable, allowing for the expansion of the donor pool. Identifying marginal livers relies in part on surgeon experience and holistic donor assessment [72,74]. Marginal livers require careful evaluation as they are associated with worse outcomes if mismatched.

Advances in hepatic disease care have acted as a major gamechanger in expanding the liver graft pool. With successful HCV treatment, previously unsuitable grafts (such as those from elderly donors or with steatosis) can now be considered for recipients who have zero viral load or those whose conditions have improved [72]. Additionally, technologies like hypothermic machine perfusion have enabled the prolongation of cold ischemia time, thus enhancing the preservation of grafts that would previously have been discarded [75]. In light of these changes, the once-considered “marginal grafts” are progressively becoming a new standard, revolutionizing graft assessment and allowing for consideration of otherwise discarded or declined livers for the best-matched recipient [76]. These include organs form steatotic organs, older donors, or those with prolonged cold ischemia times. However, due to their suboptimal physiologic state, these grafts are associated with higher risks of rejection, early graft dysfunction, and poor survival outcomes [77,78]. Due to their higher susceptibility to ischemia-reperfusion injury, which may contribute to primary non-function, close perioperative monitoring is warranted [78,79]. The long-term viability of such livers also remains questionable as a higher incidence of reduced graft survival, biliary complications, and chronic rejection has been noted in recipients compared to standard-criteria grafts [79]. The use of normothermic machine perfusion (NMP) has played a role in improving marginal graft viability; however, the risks of graft loss and early post-transplant complications need to be considered [80].

Optimizing recipient selection criteria juxtaposed with recipient waitlist prioritization can improve outcomes and provide better distribution of limited healthcare resources. Table 2 summarizes ways in which unconventional donor organs may be used for certain recipients, helping enlarge donor organ supply.

This shift in clinical attitude is also underpinned by a growing recognition that appropriate D-R matching can lead to favorable transplant outcomes, even when using extended-criteria donors. For example, prioritizing low-MELD HCC recipients for marginal grafts has been associated with better patient and graft survival rates [76,81]. This recognition has encouraged some regions, like Italy, to adopt flexible allocation systems that incorporate factors beyond the MELD score to improve D-R matching [82]. Such strategies help reduce waiting list mortality by matching patients who are most likely to benefit from transplantation with appropriate grafts, thereby improving survival rates and the equitable distribution of healthcare resources [72]. Despite ongoing debates about the risks of using marginal grafts, particularly regarding the transmission of infectious diseases or malignancy, the development of standardized guidelines for their use is seen as crucial for improving transplantation outcomes.

In conclusion, expanding the donor pool through the use of marginal grafts is an effective strategy to address the growing organ demand. The key to implementing this lies in improved D-R matching and a better understanding of donor risk factors, which help ensure better outcomes [83]. An ideal prioritization system for organ allocation should effectively evaluate patients and diseases, prioritizing those with the highest mortality risk while predicting the best post-transplant survival. In D-R matching, different variables—donor, recipient, and logistics—are combined to obtain two possible outcomes: graft survival or graft loss at various endpoints, with three and twelve months being the most commonly used timeframes [83]. Various scoring systems discussed subsequently have been developed to ensure suitable donor-recipient matching. Table 2 highlights ways to expand the donor pool.

**Table 2 biomedicines-13-00494-t002:** Considerations for Unconventional donors: Suggested means to expand the donor pool.

**Donor Factor**	**Implications**	**Potential for Use**
Hepatic Steatosis	Decreased mitochondrial membrane potential. Enhanced cell damage in cold ischemia and ischemic reperfusion injury. Graft rejection or dysfunction [84].	Severe steatosis graft is generally rejected. Grafts with moderate steatosis should be considered in combination with other risk factors, such as advanced donor age and prolonged CIT [11].
HCV positive donors	Fibrosis. Directly acting antivirals have changed the landscape of the management of HCV-infected patients [85].	If there is no significant donor fibrosis, good outcomes can be achieved from HCV antibody-positive allografts. Such organs should not be automatically rejected [85,86]. Numerous reports of successful outcomes for an HCV antibody-positive recipient exist [87,88].
Donors with HbcAb	Suboptimal graft quality, poor outcomes. Risk of reactivation and uncontrolled replication due to immunosuppression [89].	With nucleoside analogs and hepatitis B immunoglobulin, there are encouraging results with such donors. HBcAb-positive status should not be the only reason to discard a donor liver [90].
Donors with Bacteraemia and Infections	Lower graft survival [13] Caution needs to be exercised with septic donors. Some transplant centers have been using DCD from bacteremic donors with good outcomes [91].	The incidence of infection transmission is low. The source of sepsis should be remote from the liver, the donors should be under appropriate and sensitive antibiotic cover, ideally for 24–48 h, and organs from donors with multi-drug-resistant sepsis must be avoided [92].
COVID-19 Donors	Given the COVID-19 pandemic, testing for COVID-19 in the donor is standard practice. Bloodstream-related transmission remains questionable.	No evidence-based guidelines exist. Transplantation of organs other than the lungs seems to be a safe practice with a low risk of transmission. Consider donors with low viral replication (Ct > 30) at procurement [93].
Donors with HIV infections-to recipients living with HIV	Transmission of drug resistance and HIV superinfection in recipients. Data on such donors is scarce [11].	Banned in America until 2013, after the passage of the HIV Organ Policy Equity Act, allowing donations from HIV-positive donors to HIV-positive recipients. Favorable outcomes in kidney transplants [94].
Donors with Malignancy	Metastatic malignancy—excluded due to risk of tumor transmission. Exceptions of this are donors with nonmelanoma skin cancer and low-grade primary CNS tumors (grade I or II) [95].	Reports show a very low risk of transmission of donor-derived malignancies by the donor organ. The potential risks and benefits should be weighed against the risks of waiting time and the urgency of the transplant. The risk stratification is not absolute [96].
Donors with Blunt liver Trauma	Poor graft function Specific liver trauma management during transplantation [97]	The French registry data reported 142 LTs from donors with recent liver trauma. The one-year overall and graft survival rates were 85% and 81%, respectively, while the 5-year rates were 77% and 72%. This suggests that donors with recent liver trauma may be safe and acceptable [97].

### 3.4. Donor Risk Scoring Systems: Donor Risk Index (DRI) and Model for End-Stage Liver Disease (MELD)

Assessment of early allograft function and risk of graft failure continue to be significant challenges in liver transplantation. Various scoring systems are used to assess donor risk and predict graft function and patient survival. Two of the most commonly discussed models are the Donor Risk Index (DRI) and the Model for End-Stage Liver Disease (MELD). Although these scoring systems are helpful, they do not always correlate well with long-term graft survival or postoperative graft function [98]. The Donor Risk Index (DRI) is a scoring system used to evaluate liver quality by considering factors like the donor’s age, the presence of liver steatosis, and the duration of cold ischemia time. Measures such as DRI are currently rarely used due to inaccuracies in predicting post-transplant survival, not accounting for relevant donor factors [99]. Studies have shown that the DRI lacks high prognostic value, particularly in the case of extended criteria donors (marginal livers), and is often not used in clinical practice due to its complexity and low predictive accuracy [100,101]. MELD Score is evidence-based and objective, has reduced mortality by decreasing waiting list time, and allows stratification of patients, but it fails to address all aspects of liver disease severity [1]. For instance, the health status of patients with specific liver conditions may not be reflected, like in cases of hepatocellular carcinoma or those with minimal liver function still registered as having low MELD scores. Additionally, important factors like age, the frailty of the donor, or the presence of other critical illnesses, which affect post-transplant survival, are not accounted for in MELD [1]. To enhance MELD’s efficacy, such factors would need consideration.

### 3.5. L-GrAFT Score for Post-Operative Graft Function

New Scoring models have been introduced to improve the evaluation of graft function and predict early graft failure after transplantation. Two examples of significant risk-scoring models discussed in JAMA are the Liver Graft Assessment Following Transplantation (L-GrAFT) and the Early Allograft Failure Simplified Estimation (EASE) score. A retrospective single-center investigation of 2008 recipient livers demonstrated that L-GrAFT risk score may be a highly accurate clinical endpoint that has the potential to standardize grading of allograft function in the first 3 months postoperatively [102]. With an outstanding C static of 0.85, this scoring system has been shown to be more accurate than the EAD definition (C statistic, 0.68; *p* < 0.001) and the Model for Early Allograft Function (MEAF) score (C statistic, 0.70; *p* < 0.001). Other studies, including a multicenter validation study, confirm this finding in patients of European, American, and Asian origin and strongly support L-GrAFT’s validity and accuracy [73,102,103]. This score is calculated using post-transplant laboratory studies, namely serum aminotransferases, total bilirubin levels, INR, and platelet count for 10 days postoperatively. The L-GrAFT model is unique due to its ability to inform about the rate of change (slope) in the factors measured, as well as being the first score to take into account postoperative platelet counts, which have been known to be associated with delayed liver function. Moreover, it was found that the aspartate transaminase (AST) or bilirubin over 10 days was the strongest determinant of graft failure [102]. Beyond this application, the L-GrAFT score may serve as an excellent and highly accurate clinical endpoint to evaluate the impact and efficacy of liver-related surgical interventions.

### 3.6. e-GLR Score Predicts Early Graft Loss in Adult Live-Donor Liver Transplantation

A 2023 study discussed the e-GLR (early graft loss risk) score, a model created using key risk factors identified through a comprehensive analysis of patients undergoing LDLT [104]. This score is based on donor age, estimated graft weight, and pre-op MELD score of the recipient. The study involved 387 recipients. Through logistic regression analysis, the team validated that the e-GLR model effectively predicts early graft loss, with a C-statistic indicating high accuracy. The researchers derived the e-GLR score formula, aiming to improve preoperative decision-making and matching in LDLT. The model’s strength lies in its reliance on preoperative factors, enabling it to inform critical decisions in donor-recipient matching and aid in the selection of candidates for transplant. The results suggest that using the e-GLR score can assist surgeons in balancing donor safety with patient prognosis, especially for high-risk candidates. In summary, graft survival and immediate postoperative results can be greatly impacted by early graft dysfunction, a common and serious condition that occurs after LDLT [104]. Many definitions have been proposed for graft dysfunction by pioneers in the field like Olthoff, Nanashima, and Dhillo et al., among others [10]. Simply put, it is the failure of proper graft function, frequently detected by or linked to LFT abnormalities. Postoperative LFTs are also used to assess graft health, but early graft dysfunction remains one of the most critical issues in predicting patient survival. The L-GrAFT and e-GLR scores are among the promising new tools for addressing these challenges. Such prognostic scores are frequently suggested based on the recipient’s preoperative MELD score, donor age, and graft weight [105,106]. They facilitate easy and reliable donor-recipient matching, evaluation, and prognostication. L-GrAFT and e-GLR may offer superior predictive power in post-transplant graft function, influencing both graft survival and recipient outcomes. Further refinements in these models could provide a more comprehensive and personalized approach to liver transplantation. Table 3 describes other scores used for graft evaluation and prognosis prediction.

## 4. Organ Procurement and Transplantation Stage

### 4.1. Ex Vivo Methods for Graft Quality Assessment

Proper preservation and assessment of grafts are critical to the success of transplantation, especially when using DCD (Donation after Circulatory Death) livers, which are known to be at higher risk for complications such as ischemia-reperfusion injury (IRI) and IC, which frequently results in graft failure or patient death [11]. Advances in ex vivo machine perfusion have shown great promise in improving risk prognosis, decreasing complications, and possibly even mitigating them [112]. Static Cold Storage (SCS), Normothermic Machine Perfusion (NMP), and Hypothermic Machine Perfusion (HMP) are three techniques used to preserve donor livers in transplantation. Additionally, Normothermic Regional Perfusion (NRP) is an in vivo perfusion method that is gaining popularity and prompts ethical debate [1]. Each method has its own advantages and disadvantages, impacting the success and viability of liver transplants.

#### 4.1.1. Static Cold Storage (SCS)

SCS is an older preservation method that involves storing the liver at 4 °C in a cold preservation solution, which slows cellular metabolism, reduces oxygen requirements, and helps prevent immediate post-transplant cellular damage [113]. SCS is simpler, less costly, and less complex, making it the standard method for graft preservation. The main weaknesses of SCS are the higher risk of IRI, especially in marginal or extended-criteria donor livers, and the limited preservation window (typically 12 h), after which organ viability significantly drops. In summary, while SCS remains the standard due to its cost-effectiveness and ease of use, NMP provides advantages in preserving organ function, quality, and viability, especially for marginal grafts. The real-time organ assessment possible during NMP can help guide decisions about whether a graft is suitable for transplant, potentially improving outcomes. If the barriers of cost and practical complexity can be overcome, NMP could become a widely adopted key method for improving the viability and success of liver transplants, especially for higher-risk donors [113].

#### 4.1.2. Normothermic Perfusion (NMP)

Normothermic perfusion is a technique that preserves the liver at body temperature (around 37 °C) by pumping oxygenated blood or a blood substitute through the organ [113]. This maintains a near-physiological cellular metabolism, preserves metabolic activity, and reduces the risk of IRI, which can occur when blood flow is reintroduced after a period without oxygen. It allows for real-time assessment of liver function before transplantation. Functional evaluation of bile production, glucose metabolism, lactate clearance, and vascular resistance offer valuable insights into the graft’s function [113]. Benefits of normothermic perfusion include improved viability of marginal livers, which may not withstand traditional preservation, reduction of IRI incidence, and the potential to extend preservation times, providing flexibility in scheduling the transplant [114]. Limitations include higher costs, the need for specialized equipment, and logistical complexity when compared to traditional methods [115].

#### 4.1.3. Hypothermic Machine Perfusion (HMP)

Introduced in 2010, hypothermic oxygenated machine perfusion (HOPE) has served as a clinical milestone in improving graft quality assessment and preservation and continues to be used extensively today [116]. HOPE operates by perfusing the liver at low flow values of around 20 mL/min with new oxygenated perfusion fluid at 4–10 degrees Celsius during the back-table procedure [117]. The minimal perfusion time for livers to remain viable is 1 h [115]. The aim is to shift metabolism from anaerobic to aerobic whilst removing waste products and microthrombi and protecting the graft from oxidative stress. Equally important, the elimination of these waste products, such as succinates, leads to enhanced restoration of mitochondrial function [117]. The main benefit of HOPE is protection from IRI and decreasing biliary complications compared to grafts exposed to SCS [118]. As a result, better graft function is achieved, with improved short and long-term outcomes. HOPE may also improve the utilization of DCD grafts. Another advantage is its simple initiation method, making it implemented in many worldwide Liver transplant centers [119]. A study by Horne et al. demonstrated that the use of HOPE resulted in a significant decrease in the incidence of post-reperfusion syndrome compared to grafts in the SCS group [119]. An increasing number of clinical and experimental findings support the use of machine perfusion, demonstrating a positive impact on post-operative graft function, graft acceptance rate, as well as patient and graft survival [115,116,118,120,121,122]. Hemodynamic instabilities were also less frequent. Consequently, hypopotassemia may be a complication of HOPE due to the elimination of vasodilatory and inflammatory mediators, necessitating potassium substitution in many cases [119]. Researchers predict that developments in extracorporeal normothermic or hypothermic machine perfusion devices may enhance the assessment of moderately and severely steatotic grafts prior to implantation in the near future. This may be performed through perfusate and bile composition analysis, as well as by examining the vascular flow and macroscopic appearance. Developments may allow for the creation of NMP-based defatting protocols to aid in the expansion of the donor bank [47].

#### 4.1.4. In Vivo Perfusion Method: Normothermic Regional Perfusion (NRP)

NRP is a type of in situ extracorporeal membrane oxygenation method that utilizes a mechanical circulatory device to perfuse organs with warm oxygenated blood [74]. Several technical variants and circuit designs exist, enabling NRP to support abdominal as well as thoracic organs, even simultaneously. It is hypothesized that NRP may upregulate cellular defense mechanisms, although the precise actions and effects of this process remain largely unknown [1]. This technique is primarily of benefit for grafts from controlled DCD donors during the critical early preservation hours. By allowing viability assessment immediately after the declaration of death, NRP increases the utility of such grafts and reduces waiting list mortality. NRP also facilitates macroscopic examination, biochemical evaluation, and in situ biopsy sample collection [123]. A systemic review and meta-analysis comparing NRP with non-NRP liver transplantation revealed lower incidences of ischemic cholangiopathy, primary non-function, graft loss, recipient death, hepatic artery thrombosis, and other biliary complications [74]. The findings suggested that NRP significantly improved DCD graft quality to a degree that is comparable to gold-standard DBD grafts, thereby expanding the donor pool. Despite its potential, factors like resource demands, the need for specially trained perfusionists, logistical challenges, and increased operating times have impeded NRP’s widespread implementation [124]. Furthermore, its use stimulates ethical debate within the framework of the “dead donor rule”, which states that any vital organs can only be retrieved for transplantation once an individual’s death is declared [1,125]. Although further research on NRP remains necessary to confirm overall benefits and address challenges, it offers hope in times of organ shortage, especially from DCD donor grafts.

## 5. Post-Transplantation Assessment

### 5.1. Immediate Post-Operative Testing

After a liver transplant, immediate post-operative testing is vital to ensure that the graft is functioning properly. Various clinical parameters are monitored, such as the patient’s body temperature, urine output, regaining wakefulness—due to the graft’s ability to clear anesthetic agents from the bloodstream—normalization of respiratory effort, and change of appearance of fluid in the patient’s drain from a serosanguinous output to clear [9]. These clues provide valuable insight into the liver’s excretory function, particularly in patients with a biliary catheter, but have limitations due to lack of sensitivity and specificity. The patient may also remain asymptomatic, and patients with similar symptoms may have varying pathologies. Thus, therapeutic decisions should not be based solely on these indicators [10].

For more accurate and objective assessment, LFTs are routinely used [10]. Post-transplant evaluation in the ICU includes measuring liver enzymes, inflammatory markers, and prothrombin time, among others. Due to the physical trauma incurred by the surgical procedure, transaminase levels rise and peak within the first 2 days post-operatively before declining. GGT and ALP start to rise around the fourth day post-operatively and reach up to five times normal before decreasing. The peak in Prothrombin time should ideally be below 20 s and then progressively stabilize. Serum lactate levels reach physiological values within 12 h of the transplant in cases of adequate allograft function. Increased lactate indicates graft inability to metabolize lactate to pyruvate or overproduction in peripheral tissues due to low cardiac output. Hyperbilirubinemia is a typical phenomenon in the first few days after transplant and is not indicative of graft dysfunction. It is a result of the resorption of hematomas or residual blood in the abdomen and hemolysis of any transfused erythrocytes [10]. Nevertheless, alarming signs include the requirement of a high volume of packed red blood cells, hyperlactatemia, and consistently elevated bilirubin, all of which are significantly associated with 1-year mortality [9,10]. To reiterate, clinical assessment and laboratory tests together provide a more comprehensive picture of graft function but must be interpreted cautiously.

### 5.2. Dynamic Test as Point-of-Care Examination in Liver Transplantation

One dynamic function test is the Indocyanine Green (ICG) clearance test. This test evaluates the plasma disappearance rate (PDR) of ICG, a dye metabolized and eliminated by the liver, providing insights into hepatic perfusion and function [126]. LiMON is a portable device that measures ICG-PDR and ICG elimination through pulse dye spectrophotometry using a near-infrared wavelength finger clip sensor. ICG kinetics values, namely IGC_PDR_ plasma disappearance rate (PDR), are defined as the percentage change over time in ICG blood concentration (>18% per minute is normal), with 24% being the upper accepted limit. Graft function can be assessed by quantifying ICG clearance and ICG-PDR values after clamping arterial and portal venous inflow of resected segments. Not only is this a cost-effective method, but it also is a bedside assessment yielding ICG-PDR values in %/min and ICG retention after 15 min (R15, %). It has been deemed an invaluable tool due to its prognostic value, ease of use, and wide range of clinical applications [126]. Zarrinpar et al. reported the successful use of LiMON in quantification of liver function and its association with graft survival, being the first to identify ICG-PDR as a measure that correlates with graft survival [98]. Unlike other assays proposed in the past, this method of graft evaluation is easily performed by bedside nurses or perfusionists. This enables clinicians to monitor graft performance without the need for invasive tests [98].

The introduction of point-of-care testing (POCT) in liver transplantation, such as the use of LiMON, is revolutionizing the management of liver grafts. This method provides quick, cost-effective evaluations of liver function, which are essential for decision-making in critical post-transplant settings. By monitoring ICG clearance and PDR values, hepatic function can be assessed even in high-risk grafts, improving prognostication and reducing complications such as IRI. Table 4 provides a brief summary of the pre- and post-transplant investigations for graft assessment.

### 5.3. Post-Transplant Imaging

USG and Doppler are performed daily in the first week and twice weekly in the 2 weeks after that to monitor graft inflow hemodynamics [9]. The aim is to assess the hepatic artery for any stenosis or pseudoaneurysm, as well as examine the biliary system for any strictures, leaks, or biloma formation. The portal vein, hepatic vein, and inferior vena cava are assessed for stenosis and thrombosis [29].

### 5.4. Biomarkers and Genomic Testing

Biomarkers are increasingly studied for their potential in assessing graft function, particularly in the context of graft survival and disease recurrence [127]. The European Society for Organ Transplantation (ESOT) Consensus Statement on Biomarkers in Liver Transplantation concluded that no strong recommendations currently exist with scarce studies on this topic. One study involved metabolomic analysis, while others investigated recipient genotypes. A literature review written by an expert working group from the ESOT and International Liver Transplant Society (ILTS) investigated primary disease recurrence post-transplant. The observational non-comparative study draws attention to the role of the G-allele in position rs738409 of patatin-like phospholipase domain-containing protein 3 (PNPLA3) [127]. The presence of this allele in the recipient provided a predisposition to metabolic dysfunction-associated steatohepatitis (MASH) and is associated with increased hepatic concentration of triglycerides. Liver biopsy is usually available to a minority of such patients, and recurrent disease diagnosis is usually performed through clinical and biological criteria [127,128]. Another finding is that the altered lipid composition with levels of 10 lipids of the chemical class of phosphatidylcholines significantly decreased [127,129]. Further research is warranted before conclusions can be drawn.

Experimental studies are currently exploring Mitochondrial miR-23b-5p as a possible biomarker for assessing warm ischemia injury (WII) and as a candidate for donor liver evaluation [130]. The quality and functionality of transplanted livers can be greatly impacted by WII, which is frequently brought on by insufficient energy metabolism. Predicting the viability of marginal donor livers is difficult due to the poor accuracy of current donor assessment techniques that use traditional indicators. Research reveals that the mitochondrial apoptotic pathway’s miR-23b-5p is susceptible to ischemic stress. By suppressing ribosomal RNA in the mitochondria, specifically mt-RNR2 (16S), ischemic conditions modify mitochondrial activity and impact respiratory function. Inhibiting miR-23b-5p enhanced mitochondrial respiration and ATP production, according to laboratory studies and animal models, suggesting enhanced cell survival in hypoxic environments [130]. The clinical relevance of miR-23b-5p was further supported through liver transplantation outcomes, showing a correlation between higher miR-23b-5p levels in donor livers and elevated alanine aminotransferase (ALT) levels in recipients, a marker of liver stress [3,42,131]. High expression of miR-23b-5p was associated with reduced graft survival rates over 20 months [130]. These findings suggest that miR-23b-5p could serve as a preoperative biomarker for WII, offering a novel method for assessing donor liver quality and potentially guiding clinical decision-making in liver transplantation.

Another study explores the role of MICA (MHC class I-related chain A) in predicting graft survival and liver function post-liver transplantation [132]. In zero-hour biopsies, MICA expression outperformed other clinical indicators such as donor age, body mass index, and the Eurotransplant Donor-Risk Index (ET-DRI), demonstrating a good prognostic value for both short- and long-term liver function. Improved liver function was associated with high MICA expression at several time points, exhibiting negative relationships with bilirubin, ALT, and AST throughout a period of 3 to 36 months. Importantly, during a 10-year period, high MICA mRNA levels were also connected to longer graft survival, while low expression was linked to mortality from transplants. These results provide credence to MICA’s usage as a possible biomarker for evaluating the quality of liver grafts prior to transplantation. In summary, clinical picture, dynamic POCT, and emerging biomarkers like miR-23b-5p and MICA are emerging tools in modern liver transplantation. They assist in the evaluation of graft function in real-time, predict long-term outcomes, and help tailor patient care based on individual graft quality [132].

## 6. Emerging Technologies—Auxillary and Novel Tests

### 6.1. Artificial Intelligence (AI) and Machine Learning

Artificial Intelligence (AI) and Machine Learning (ML) are transforming liver transplantation by enhancing donor-recipient matching, predicting graft function, and assessing liver viability [133]. AI tools, particularly Artificial Neural Networks (ANNs), are being developed to analyze large datasets and improve decision-making in liver transplantations. They remove emotional biases and can be used to improve decision-making speed and accuracy [133]. Briceno et al. first applied a neural network to develop a donor-recipient allocation model (M.A.D.R.E), using various donor and recipient variables to predict graft failure at three months [134]. ANNs outperformed traditional prioritization scores such as MELD and D-MELD in donor allocation. Subsequent studies have reinforced these results, showing that neural networks excel in processing complex data patterns for better predictions [134,135].

Despite the rapid advancement of AI in liver disease management, clinical studies focused specifically on donor-recipient matching remain limited. Some research suggests that the superiority of complex deep learning systems may not hold when applied to larger databases, such as those from UNOS [42]. The field continues to seek robust, effective scoring systems applicable to extensive datasets with the aim of streamlining the allocation process and improving speed, cost-efficiency, and overall outcomes.

Machine learning algorithms have been evaluated in recent years in the area of predicting long-term mortality risk after solid organ transplants [136]. For liver transplantation, these models have been used to assess the short-term survival of both the graft and the patient after the procedure [127]. Transplant recipients can receive one-year and five-year forecasts for the four main causes of mortality at any time following transplantation [127]. Results from the research conducted by Nitski et al. indicate that machine learning could enhance post-transplant care by predicting outcomes using non-linear structures, which identify relationships between features that standard biostatistics may miss [128]. Moreover, factors such as HCV infection, donor age, rejection following transplantation, as well as post-transplant diabetes, hypertension, and adrenal insufficiency have been identified as predictive variables for graft-related mortality [129,130,131]. Despite the fact that these algorithms do not play a direct role in graft assessment, their ability to predict outcomes provides valuable information that, in the future, may be used to help investigate graft quality preoperatively [128]. Similarly, deep learning algorithms can utilize longitudinal data to continuously forecast long-term outcomes following liver transplantation, surpassing the performance of logistic regression models. Physicians could employ these algorithms during routine follow-up visits to identify recipients at risk for negative outcomes and take preventive measures [128].

Piella et al. addressed the obstacle of error-prone macroscopic inspection (for HS) with the development of LiverColor, a collaboratively created software platform that utilizes image analysis and machine learning to determine graft eligibility, largely based on its level of steatosis [42]. An internal dataset of 192 grafts was used to create and validate the classification models. LiverColor integrates liver imaging data and clinical features (such as ALT, BMI, and GGT) to determine eligibility. The algorithm, which utilizes techniques like random forests and support vector machines, is superior to subjective visual inspections by surgeons. Initial findings indicate that LiverColor outperforms the surgeon in predicting hepatic steatosis >15%, demonstrating a higher accuracy of 85% compared to 73%, precision of 92% versus 51%, and recall of 89% against 68%. The findings suggest that combining image analysis with machine learning has the potential to improve the accuracy and efficiency of liver evaluations, lessening dependence on subjective visual assessments and enhancing transplantation outcomes. In conclusion, LiverColor offers a straightforward, real-time, and relatively precise evaluation of HS [42]. Validating the model on external datasets from various centers is vital to assess its performance across different clinical settings and diverse patient groups.

Whilst AI models offer a broad range of opportunities to enhance graft assessment, it is worth mentioning that they are not without shortcomings. Data bias and lack of transparency in decision-support systems remain challenges to overcome. The use of training datasets revealed disparities leading to unequal organ allocation and under-representation of diverse graft populations [137]. The implications of biased predictions may also result in favoring certain demographic areas while depriving others and subsequently leading to suboptimal transplant outcomes [135]. This only reinforces healthcare inequalities and necessitates the development of debiasing protocols and diverse data-sourcing techniques if AI is to play an active role in the grand scheme of liver transplantation and the medical field as a whole [138]. Issues with transparency also raise ethical questions. Transparency in AI-based decision-making is crucial to ensure that final outcomes are in keeping with clinical judgments [139]. The lack of this, termed as “black box” nature, creates barriers for clinicians and patients in placing their trust and accountability in such systems [140]. For instance, clinicians must be able to access the underlying rationale for AI rejecting a particular graft to determine whether the recommendation aligns with their clinical judgment. Enhancing transparency through explainable AI (XAI) techniques is essential to bridge this gap [141]. Medico-legal and regulatory bodies have highlighted the need for clear documentation and validation of model frameworks to facilitate clinical integration and avoid possibilities of devastating outcomes [142].

### 6.2. Regenerative Medicine: Bioengineered Organs as Alternatives

The field of regenerative medicine holds significant promise in addressing organ shortage. Recent advancements in bioengineered liver grafts have been groundbreaking. One intriguing study describes the use of a decellularized pig liver scaffold to create a human-scale, bioengineered liver graft that was repopulated with primary hepatocytes and endothelial cells [3]. It was designed as a preclinical model to evaluate its use for transplantation. The multistep perfusion and pressure monitoring in the recently discovered biofabrication technique allowed for enough hepatocyte repopulation in the parenchymal space to restore liver function [132,133]. Urea secretion, coagulation factor production, and albumin production all steadily increased prior to transplantation. Genes and microRNAs linked to several metabolic processes were also upregulated [134]. Pigs with induced liver failure were used to test the graft [3]. It was found to successfully support liver function for 28 days. This is the first time a bioengineered liver in a large animal model has demonstrated such extended post-transplant survival and function. The ability to sustain liver-specific functions like albumin production and bile acid secretion marks an important step forward in creating alternative sources of liver tissue for transplantation [3].

Limitations, such as fibrosis at the site of the portal vein anastomosis, led to decreased blood flow toward the conclusion of the research period [3]. The study offers hope that bioengineered liver transplants can potentially alleviate the scarcity of donor organs and provide a regenerative treatment for end-stage liver disease, notwithstanding these obstacles. In order to increase the therapeutic relevance of bioengineered liver grafts, the researchers hope to include human liver cells made from induced pluripotent stem cells (iPSCs) in the future [135]. Such exciting studies remain in their infancy. In an era where pig-to-human heart xenotransplantation was successfully carried out, it is only fascinating to imagine how liver transplantation will be revolutionized in the years to come.

## 7. Conclusions and Future Directions

Liver transplantation is a critical yet complex procedure in modern medicine, evolving with advancements in technology, assessment techniques, and resource allocation. With liver disease becoming increasingly prevalent globally, the demand for donor organs has outpaced supply, resulting in high waitlist mortality rates. Traditional criteria for graft assessment often discard viable livers, prompting the adoption of refined tools such as dynamic tests, imaging, and biomarkers to expand the donor pool. More importance is being attached to imaging and AI in graft evaluation. AI, machine learning, and regenerative medicine are set to revolutionize liver transplantation. Their potential lies in optimizing donor-recipient matching and post-transplant care while at the same time offering platforms like LiverColor, which can provide relatively accurate and reliable liver graft evaluations [42]. In regenerative medicine, bioengineered liver grafts can be harnessed to overcome organ shortage, providing a new resource opportunity in the future. These advances highlight the growing role of technology in improving outcomes and expanding donor organ availability. They provide a wealth of hope, but much research into these techniques is warranted before authorized and standardized clinical use can be integrated into patient care.

Meanwhile, measures such as CAP and LS from FibroScan, juxtaposed with MRE and other imaging modalities, have demonstrated both accuracy and utility during the examination of grafts with HS and fibrosis, which remain significant obstacles for transplant surgeons today. Data demonstrate that these methods continue to provide superior results in comparison to liver biopsy, which is quickly coming out of favor and losing accuracy. The LiverColor application is useful in predicting HS of higher than 15%, while HAI from CT helps categorize the degree of steatosis as mild, moderate, or severe. The remaining challenges in liver transplantation are graft rejection (due to HS), preservation injury, ischemic reperfusion injury, ischaemic cholangiopathy, and recurrence of native liver disease.

In an attempt to explore ways to expand the donor pool, the concept and use of marginal grafts were also discussed. Marginal grafts are quickly becoming a standard and continue to play a crucial role in the background of organ shortage. They have broadened their definition now to include donors with Hepatitis C donors, liver trauma history, COVID-19, and Hepatitis B antibodies, among others, with successful outcomes documented in the literature [49,72]. Modern procurement techniques, such as NMT and HOPE, have facilitated the use of these grafts and hold promise for enhancing graft viability and prolonging preservation periods, avoiding graft injury, and mitigating IC [55]. They also allow improved utilization of DCD livers [55]. Flexibility around donor age, gender, and graft size has also been witnessed in recent years. Using grafts from older people has been essential to expand the donor pool, but this demands careful selection of the recipient and minimizing CIT. Although no specific limits exist regarding graft size, recipient imaging is one guiding factor, alongside considering risks and prophylaxis for portal hypertension. Donor gender remains an inconclusive variable.

While marginal grafts open doors of opportunity, they come with the unfortunate hurdle of the need to develop finely crafted and holistic donor-recipient matching and graft allocation systems to optimize graft use and minimize complications. Stepwise evaluation of the target donor is key to implementing this, mainly through consideration of case-by-case scenarios, using a multidisciplinary approach, and utilizing scoring systems effectively. Scoring systems are valuable tools for graft allocation; however, no one-size-fits-all score exists. MELD has been the standard in the US, but it lacks predictive power [1]. L-GrAFT and e-GLR show promise but require further clinical verification. All in all, despite unprecedented organ demand, continued advances in procurement, evaluation, and scoring systems offer new opportunities for patients on waitlists, signaling a transformative era in liver transplantation practices.

## Figures and Tables

**Figure 1 biomedicines-13-00494-f001:**
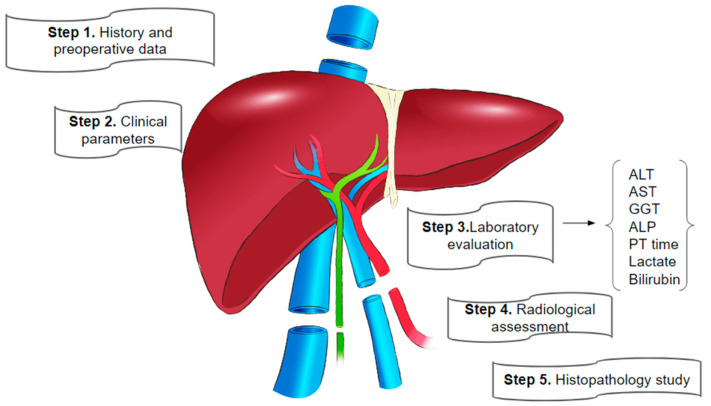
Key steps in liver graft evaluation during preparation for transplantation [10,50].

**Table 1 biomedicines-13-00494-t001:** Factors of donor and graft: literature findings within the last 5 years.

Factors	Reference	Place and Year of Publication	Findings	Current Consensus
Age	Gilbo N. et al. [64]	Belgium, 2019	Older donor grafts can be safely utilized in older recipients if other risk factors are minimized.	No donor age limit is set due to improved outcomes with older donors.
	Nakamura T. et al. [65]	Japan, 2022	Liver grafts from elderly donors show slower recovery patterns during the initial phase but eventually lead to satisfactory outcomes.	
	Maestro O. C. et al. [66]	Spain, 2022	The outcomes from grafts of nonagenarian donors are similar to those from octogenarian donors.	
Liver graft Size	Reyes J. et al. [67]	U. S., 2019	In deceased donors, the ratio of donor to recipient body surface area is an important predictor of graft survival.	The maximum implantable graft volume in cirrhotic patients is the sum of the recipient’s liver volume and the right upper abdominal cavity’s dimensions. This volume also correlates with portal hypertension severity.
	Addeo P. et al. [68]	France, 2022	Integrating donor anthropometrics with recipient imaging can enhance donor-recipient matching processes and help prevent complications.	
	Kostakis I. D. et al. [69]	U. K., 2023	A mismatch in size is linked to higher rates of portal vein thrombosis within the first three months in data obtained from 85% deceased brain-dead donors (DBD) and 15% from deceased circulatory-dead donors (DCD).	
Gender	Rustgi V. K. et al. [70]	U. S., 2022	Patients with a gender mismatch have a 6.9% higher risk of graft failure.	The impact of gender mismatch on post-transplant outcomes remains debated, requiring larger, well-calibrated studies to clarify its potential effects in liver transplantation.
	Germani G. et al. [71]	Italy, 2020	In male recipients, a mismatch in donor-recipient gender, along with the use of obese donors for female recipients, is associated with lower survival rates after liver transplantation.	

**Table 3 biomedicines-13-00494-t003:** Scores for donor-recipient matching.

Score System	Reference	Place and Year of Publication	Factors	Limitation
P-SOFT (the Preallocation score to predict Survival Outcomes Following Liver Transplant Score) and SOFT Score	Rana A. et al. [107]	U. S., 2008	Donor age BMI History of transplant Albumin levels Need for Dialysis ICU admissions MELD score Life support Encephalopathy Portal vein thrombosis Ascites	Subjective variables and complexity limit clinical use in pretransplant decisions.
D-MELD	Halldorson J. B. et al. [108]	U. S., 2009	The product of donor age and preoperative MELD score	Weak predictive power.
BAR	Dutkowski P. et al. [109]	Switzerland, 2011	Donor age Recipient age CIT Retransplantation need Life support need MELD score	Does not account for graft steatosis and suboptimal function. Inaccurate in predicting transplant survival.
ET-DRI (Eurotransplant-Donor-Risk-Index)	Braat A. E. et al. [110]	Eurotransplant region, 2012	Donor age Cause of death DCD Split liver grafts Organ location CIT Rescue allocation GGT levels	Limited effectiveness in predicting early outcomes after liver transplantation.
ISO (Italian Score for Organ allocation)	Cillo U. et al. [111]	Italy, 2015	MELD score Urgency HCC	Needs prospective validation to confirm superiority over MELD score.

**Table 4 biomedicines-13-00494-t004:** Graft assessment methods before and after transplantation.

Pre-Transplant	Post-Transplant
Biochemical (static) tests—LFT	LFT, Routine post-operative blood tests
Dynamic tests—ICG	Ultrasonography—Doppler
Imaging-Ultrasonography, Fibroscan, CT, MRI	Clinical assessment
Liver Biopsy, with the role of mass spectrometry	ICG-PDR/LiMON as a point-of-care test
Macroscopic inspection by Surgeon	Biomarkers and AI-guided imaging
Histopathological evaluation	
Donor risk scoring	
AI-guided technologies—LiverColor app and others

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
