# Peer review of "Perspectives and Tools in Liver Graft Assessment: A Transformative Era in Liver Transplantation"

_biomedicines, 2025, doi:10.3390/biomedicines13020494_

Round 1

Reviewer 1 Report (Previous Reviewer 1)

Comments and Suggestions for Authors

The authors addressed systematically the problems outlined in the previous review. The article is now well structured and offers a comprehensive outline of current strategies for evaluating liver grafts for beginners. Some minor English mistakes are found throughout the text, but this can be addressed during the final editing process. 

Author Response

Thank you for your comments. We have made grammar edits and highlighted them in green. For example, missing articles were inserted, numbers written out and word choice improved. Thank you. 

Reviewer 2 Report (New Reviewer)

Comments and Suggestions for Authors

I have read and analyzed the manuscript from Safi and coauthors. In my opinion, the problem of liver transplantation today is one of the most significant in modern medicine and this review obligatory should be considered for publication. I think that the presented review is complete and comprehensive, nevertheless I have a couple of small questions for enhancement of the presented review.

1.Which is the place of ultrasound imaging and echogenicity determination in the pre-donation assessment phase? It should be added in the manuscript.

2.What about the using of liver metabolome in the pre-donation assessment phase? Do authors have some information about mass-spectrometry using in liver quality assessment?

3.Authors in the end of their comprehensive review should add final block-scheme of pre-donation liver assessment and post-transplantation liver assessment methods. In my opinion, this review requires any summarizing as a circumstantial image or scheme.

Author Response

Reviewer 3 Report (New Reviewer)

Comments and Suggestions for Authors

This review article is supposed to talk about liver graft assessment. This could be done under 3 separate sections:

1) Assessment of deceased donor graft

2) Assessment of a living donor graft

3) Post-operative graft assessment

However, the article is very incoherent in its flow with discussion all over the place, random, jumping from topic to topic.

Although graft assessment for living donor and deceased donor transplants are done quite differently, in this article they have been used interchangeably, which does not give a clear message to the reader

Based on the title, a transformative era in graft assessment is supposed to be discussed, although there is little to offer the reader in that. In fact, the authors have missed an opportunity to discuss graft assessment on nRP or ex-vivo machine perfusion.

Round 2

Reviewer 2 Report (New Reviewer)

Comments and Suggestions for Authors

Many thanks to authors or the corrections, in my opinion the manuscript can be accepted for publication

Author Response

We strongly agree with your comment and appreciate your opinion. We have altered the colors and highlights of the text to match the rest of the manuscript and attempted to organize the tables better. 

Please kindly note that only the first and last (corresponding) author emails will be included on the final manuscript. 

Please find the latest manuscript copy attached below. Thank you very much once again. 

This manuscript is a resubmission of an earlier submission. The following is a list of the peer review reports and author responses from that submission.

Round 1

Reviewer 1 Report

Comments and Suggestions for Authors

In their review, “Overview of available options for assessing the condition and functionality of the transplanted liver - where are we at this moment?”, the authors tried to summerize evidence on current strategies to assess graft function in liver transplatation. However, their review lacks some very important information and does not take into account important progresses made during the last decade.

Major issues:

-              Most of the information is reduntant and can be found in multiple places throughout the manuscript 

-              hypothermic oxygenated machine perfusion has become extensively used to improve graft quality and this has not been taken into account 

-              the problem of liver graft quality should be divided in DCD and non-DCD donors

Minor issues:

-              The references are not cited in the order they apear in the text

-              Some important references are missing to some of the information provided throughout the manuscript (e.g. introduction – “Static liver tests standard for preoperative surgical planning include aspartate transaminase, alanine transaminase, and the Model for End-stage Liver Disease (MELD) to measure the liver’s functional reserve.” – to my knowledge MELD has not been validated to assess donor’s liver). 

-              “This is problematic as liver biopsy and rapid histological diagnosis is not routinely done for marginal livers (livers from higher risk donors)”. Please provide a reference as biopsy is standard of care for marginal livers in most centres. 

Comments on the Quality of English Language

The text needs extensive english editing to make it more fluent and easy to read.

Reviewer 2 Report

Comments and Suggestions for Authors

The authors concluded that "although further studies will be required to address the conclusions of this review, it remains a common notion that no single static or dynamic test correlates with pre-transplant graft function and survival. It also remains the fact that multiple interrelated donor factors, as well as recipient factors, make the tasks of predicting graft function and survival, monitoring the postoperative course and donor-recipient matching a challenge [49]. In the light of worldwide organ shortage, this challenge remains an issue of importance for all researchers in the field. Other hurdles impeding progress are the inability to standardize the transplant surgeon’s graft assessment and the inability to initiate such assessment without sitting in an operating room and performing procurement procedures [49]. 

Future hopes are to develop a validated pre-procurement assessment of liver graft function that would optimise donor selection and result in greater efficiency."

They say that this validated assessment of liver graft function before procurement will be developed to optimise donor selection and result in greater efficiency. However, they do not specify whether they are going to be the ones to develop it and what elements, questions or values it should have to be a “star” tool that would solve all the problems posed by the other tools analyzed.  

Comments on the Quality of English Language

Please, also moderate editing of the English language required.
